# Water-Soluble Pd Nanoparticles for the Anti-Markovnikov Oxidation of Allyl Benzene in Water

**DOI:** 10.3390/nano13020348

**Published:** 2023-01-14

**Authors:** Edwin Avila, Christos Nixarlidis, Young-Seok Shon

**Affiliations:** Department of Chemistry and Biochemistry, California State University Long Beach, 1250 Bellflower Blvd., Long Beach, CA 90840, USA

**Keywords:** nanoparticle, catalysis, semiheterogeneous, oxidation, anti-Markovnikov, green catalysis, biphasic catalysis

## Abstract

The catalytic activity and selectivity of two different water-soluble palladium nanoparticles capped with 5-(trimethylammonio)pentanethiolate and 6-(carboxylate)hexanethiolate ligands are investigated using the catalytic reaction of allyl benzene. The results show that the regioselective transformation of allyl benzene to 3-phenylpropanal occurs at room temperature and under atmospheric pressure in neat water via a Tsuji–Wacker type oxidation. Conventionally, the Tsuji–Wacker oxidation promotes the Markovnikov oxidation of terminal alkenes to their respective ketones in the presence of dioxygen. Water-soluble Pd nanoparticles, however, catalyze the anti-Markovnikov oxidation of allyl benzene to 3-phenylpropanal in up to 83% yields. Catalytic results of other aromatic alkenes suggest that the presence of benzylic hydrogen is a key to the formation of a p-allyl Pd intermediate and the anti-Markovnikov addition of H_2_O. The subsequent b-H elimination and tautomerization contribute to the formation of aldehyde products. Water-soluble Pd nanoparticles are characterized using nuclear magnetic resonance (NMR), UV–vis spectroscopy, thermogravimetric analysis (TGA), and transmission electron microscopy (TEM). Catalysis results are examined using ^1^H NMR and/or GC-MS analyses of isolated reaction mixtures.

## 1. Introduction

The high surface area-to-volume ratio and the increased presence of active sites with low coordination numbers make metal nanoparticles excellent materials for chemical catalysis [1,2]. These characteristics consequently lead to a greater degree of reactivity since the substrate and the catalyst can interact to a larger extent. However, metal nanoparticles have a tendency to aggregate during and after catalytic reactions leading to deteriorations in activity and/or recyclability. The use of a support such as silica, metal oxide, and carbonic materials to bind the metal nanoparticles (Au, Pd, Pt, etc.) physically or chemically has been a popular strategy to alleviate the stability issues associated with metal nanoparticles [3,4,5]. Although these support systems have often proven to be more effective for stabilizing metal nanoparticles than their counterpart, organic ligand stabilization, they have comparably exhibited diminished reactivity and selectivity [6]. On the contrary, ligand-stabilized colloidal nanoparticles, which are semiheterogeneous in nature, can benefit from the characteristics of both homogeneous and heterogeneous systems. Specifically, they can be dispersed in aqueous or organic solvents allowing for homogeneous reactivity while also retaining effective separability common to heterogeneous systems [7,8,9,10,11].

Ligand-capped metal nanoparticles have shown a profound dependence on their surface compositions for exhibiting their degree of reactivity and selectivity [7,8,9,10,11,12,13]. A key aspect for achieving the desired selectivity often involves the partial poisoning of the catalyst surface with the ligand of choice. Our previous studies have shown that by employing various alkanethiosulfate preligands the appropriate balance between reactivity and selectivity of Pd nanoparticles can be achieved for organic transformations [14,15,16,17]. The slower passivation kinetics of ionic head groups in this alternate ligand have allowed for the control of surface ligand density for Pd nanoparticles. The ligand structure, conformation, and surface density on the metal catalyst were also crucial in obtaining the desired activity/selectivity for different catalysis reactions [9,18,19]. These Pd nanoparticles synthesized using the thiosulfate protocol have been successfully employed in several organic reactions including, for example, the chemoselective hydrogenation of styrene derivatives, the stereoselective hydrogenation of allenes, the regioselective hydrogenation of dienes, and the regioselective isomerization of allylic alcohols [9,20,21].

The Tsuji–Wacker oxidation involves the selective conversion of terminal alkenes to methyl ketones following the Markovnikov addition of the oxidant about the more substituted carbon center (Figure 1) [22,23,24]. Increasing the versatility of this reaction required, however, the addition of cosolvent such as dimethylformamide (DMF) to address the insolubility issue observed for larger olefinic substrates [25,26]. Though versatile in its catalytic utility, the recent focus of the chemical community for the Tsuji–Wacker type oxidation has been on the selective generation of aldehydes via the anti-Markovnikov oxidation [27,28,29,30,31,32,33,34,35]. This is partly due to the functional handle that aldehydes provide for further synthetic modification [36,37]. The regioselective oxidation of terminal olefins via the anti-Markovnikov addition of water about the less substituted carbon center is of substantial importance to the catalysis community since it provides an appealing alternative to conventional methods such as hydroformylation and hydroboration/oxidation [38,39].

Transition metal catalysts have proven to be rather effective for the anti-Markovnikov Tsuji–Wacker oxidation of terminal alkenes. The regioselectivity for the addition of the oxidant is contingent upon several factors including the choice of solvents, absence of reoxidant, strategic functional group presence, bound ligand, and more [28,29,30]. For example, the presence of directing groups such as alcohols, amines, phosphonates, and siloxy groups is postulated to allow for chelation to the metal center, which promotes the anti-Markovnikov addition about the less hindered carbon [29,30,31,32,33]. The absence of a reoxidant, which has served to complete the catalytic process by oxidizing Pd^0^ back to Pd^II^ has also allowed for the selective production of the aldehyde in a 9.8:1 ratio [30]. Recently, Grubbs and coworkers have demonstrated the nitrite-modified Tsuji–Wacker oxidation which uses AgNO_3_ as a co-catalyst. This has resulted in the formation of aldehydes up to 70% with 90% selectivity [31,32]. Despite these results, it is important to note that these reactions require the use of expensive cocatalysts (AgNO_3_), reagents (t-BuOH, MeNO_2_, etc.), unsafe/problematic solvents (DMF), and/or stoichiometric amounts of catalysts/oxidant to proceed successfully [28,29,30,31,32]. The conditions for this reaction have generally proven to be unappealing for industries and chemists alike.

Herein, we highlight an alternative approach that encompasses the use of water-soluble Pd nanoparticles (PdNPs) for the anti-Markovnikov oxidation of terminal alkenes under mild and greener conditions. In addition to the previously reported carboxylate-terminated Pd nanoparticles [40,41,42], ammonium-functionalized ligand-capped Pd nanoparticles that are soluble in water are applied as catalysts for biphasic oxidation of allyl benzene to 3-phenylpropanal (Figure 2). Notably, no cosolvents or additional reagents besides H_2_ (g) were needed in this process for successful conversion.

## 2. Materials and Methods

The following reagents were purchased from the indicated suppliers and used as received. Tetra-*n*-octylammonium bromide (TOAB), sodium borohydride (NaBH_4_), and potassium tetrachloropalladate (II) (K_2_PdCl_4_) were obtained from Acros. 6-Bromohexanoic acid (C_6_H_11_O_2_Br), (5-bromopentyl)trimethylammonium bromide, allyl benzene, styrene, 4-phenylbut-1-ene, and sodium thiosulfate pentahydrate were obtained from Sigma-Aldrich. Ethanol, methanol, acetone, dichloromethane, chloroform, acetonitrile, and toluene were obtained from Fisher Scientific. Deuterium oxide (D_2_O) and chloroform-d (CDCl_3_) were purchased from Cambridge Isotope Laboratories, Inc. Water was purified using a Millipore Sigma Simplicity Ultrapure Water System.

**Synthesis of S-6-(carboxylate)hexyl thiosulfate.** The synthesis of sodium ω-carboxylate-S-hexyl thiosulfate was achieved following a previously published method [40,41,42]. In a 500 mL round bottom flask equipped with a magnetic stir bar and a condenser, 25 mmol of 6-bromohexanoic acid and 25 mmol of sodium thiosulfate pentahydrate were mixed in 100 mL of solvent (50:50 ethanol and water). The flask containing the mixture was then refluxed while stirring for 3 h. After reflux, the solution was allowed to cool to room temperature after which the solvents were removed via rotary evaporation. The crude product was dissolved in hot ethanol, cooled to 0 °C, and allowed to recrystallize overnight. The final product was filtered and then washed repeatedly with cold ethanol. The white crystalline solid was collected in a vial and stored under a vacuum for future use. The ^1^H NMR result of ω-carboxylate-*S*-hexyl thiosulfate sodium salt ligand is shown in Appendix A.

**Synthesis of S-(5-trimethylammonio)pentyl thiosulfate.** In a 50 mL round bottom flask equipped with a magnetic stir bar and a condenser, 8.65 mmol of N-(5-bromopentyl)trimethylammonium bromide and 9.08 mmol of sodium thiosulfate pentahydrate were mixed in 10 mL of water. The reaction mixture was heated and maintained between 70 °C and 80 °C in an oil bath for 6 h while stirring. Once complete, the solution was allowed to cool overnight, and the product was precipitated while stirring gently. The final product was further cooled to 0 °C, filtered, and then washed repeatedly with cold H_2_O. The white crystalline solid was collected in a vial and stored under a vacuum for future use. The ^1^H NMR result of S-(5-trimethylammonio)pentyl thiosulfate sodium salt ligand is shown in Appendix A.

**Synthesis of 6-(carboxylate)hexanethiolate- and 5-(trimethylammonio)pentanethiolate-capped Pd nanoparticles.** K_2_PdCl_4_ (0.40 mmol) was dissolved in 12 mL of nanopure water and TOAB (2.0 mmol) was dissolved in 25 mL of toluene. Both solutions were mixed and continuously stirred until the organic layer turned dark orange and the aqueous layer became clear, indicating the completion of the phase transfer of PdCl_4_^2−^. The aqueous layer was discarded using a separatory funnel, and the organic layer was placed in a 500 mL round-bottom flask. S-(5-Trimethylammonio)pentyl thiosulfate sodium salt (0.80 mmol) dissolved in 10 mL of 25% methanol was added to the organic layer; additional TOAB (2.0 mmol) was then added to the reaction flask. The reaction mixture was continuously stirred for 15 min. Afterward, NaBH_4_ (8.0 mmol) in 7 mL of nanopure water was vortexed for ~15 s and delivered to the reaction flask dropwise over the span of 60 s. Consequently, the solution darkened immediately, indicating the formation of nanoparticles. While stirring for 3 h, the PdNPs were transferred into the aqueous layer. The organic layer was then discarded using a separatory funnel, and the solvent was removed via rotary evaporation. The resulting crude nanoparticles were partially suspended in 25 mL acetonitrile/H_2_O (90:10) (methanol for 6-(carboxylate)hexanethiolate-capped PdNPs) and centrifuged at 4900 RPM. The precipitated nanoparticles were then washed twice with acetonitrile (methanol and ethanol for 6-(carboxylate)hexanethiolate-capped PdNPs). The washing procedure included sonicating the nanoparticles in the respective solvent and centrifuging at 4900 RPM. The resulting nanoparticles were dissolved in water and dialyzed overnight (~14 h). After dialysis, the water was removed via rotary evaporation and the resulting nanoparticles were dried in a vacuum overnight at a pressure of 25 psi.

**Characterization of palladium nanoparticles and ligand precursors.** The ^1^H NMR spectra were taken using a 400 MHz spectrometer (Bruker Fourier, Billerica, MA, USA). NMR data were analyzed via MestreNova software. D_2_O was used as the solvent of choice for both ligands and nanoparticles. The UV–vis spectra of the synthesized PdNPs were obtained using a Shimadzu UV-2450 spectrometer with nanopure water serving as the solvent. Data were analyzed via UV probe software. Thermogravimetric analysis (TGA) was performed using TA Instruments SDT Q600. Approximately 5 mg of PdNP sample were placed in an aluminum pan and loaded into the instrument. Analysis was conducted under the following parameters: temperature ramp rate of furnace set at 20 °C/min and max temperature set to 600 °C. Data were analyzed via Trios software. Transmission electron microscopy (TEM) images were obtained by using a 1200 EX II electron microscope (JEOL USA, Peabody, MA, USA). The preparation of the TEM sample was achieved by diluting PdNPs to 0.002 mg/mL in methanol solvent. The samples were then deposited onto a 400-mesh standard carbon-coated copper grid and allowed to dry in the air for approximately 30 min. The images for the particle size distribution were analyzed via Scion Image Beta Release 2.

**Catalytic assays.** Catalysis experiments were performed by adding 10 mol% PdNP catalysts (0.025 mmol based on Pd) dissolved in 2 mL nanopure water into a 100 mL glass round-bottom flask equipped with a rubber stopper and a magnetic stirring bar. The reaction mixture was purged with hydrogen gas for 10 min. After the influx of H_2_ gas, the substrate (0.25 mmol) was injected into the sealed reaction flask. The reaction was continuously stirred at room temperature for 24 h. Reaction products were extracted twice by adding 1 mL of CH_2_Cl_2_. Purification of organic products from PdNP residues and water was done by filtration through a Pasteur pipette silica gel column chromatography using CH_2_Cl_2_ as eluent. The isolated products were analyzed via ^1^H NMR and/or GC/MS.

## 3. Results and Discussion

### 3.1. Synthesis of Water-Soluble Palladium Nanoparticles

Quaternary ammonium- and carboxylate-functionalized Pd nanoparticles were successfully synthesized via the two-phase thiosulfate synthesis, a modified Brust–Schiffrin method [40]. The previous report confirmed that the biphasic synthetic condition prevents the hydrolysis of the sulfite moiety, and the thiosulfate preligands decrease the rate of surface passivation allowing the formation of nanoparticles with higher catalytic activities [9]. A scheme outlining this thiosulfate process is depicted below (Figure 3).

Briefly, the synthesis of these nanoparticles began with the phase transfer of the Pd^2+^ salt to the organic layer. This involved the use of a phase transfer reagent, tetraoctylammonium bromide (TOAB). After the phase transfer, the ligand precursor was added to the solution and transferred to the organic layer via additional TOAB. The Pd^2+^ salt was then effectively reduced to Pd^0^ via the addition of sodium borohydride (NaBH_4_). The reduction of palladium in the presence of capping ligands allowed for the formation of appropriately sized nanoparticles via nucleation, growth, and passivation of PdNPs.

### 3.2. Characterization of Palladium Nanoparticles and Ligand Precursors

The ^1^H NMR spectrum for the nanoparticles synthesized with the sodium ω-carboxylate-S-hexyl thiosulfate ligand precursor is shown in Appendix A, which agrees with the previously published results [40]. The ^1^H NMR spectrum for the newly synthesized PdNPs prepared from S-(5-trimethylammonio)pentyl thiosulfate ligand precursor is shown in Appendix A. The absence of methylene proton signals (α and β CH_2_ signals to the S) indicates the successful covalent binding of the thiolate group to the metal surface. This is also confirmed by the substantial broadening of visible peaks.

To further confirm the successful formation of nanoparticles, the UV–vis spectra of both 5-(trimethylammonio)pentanethiolate-capped PdNPs (C5-PdNP) and 6-(carboxylate)hexanethiolate-capped PdNPs (C6-PdNP) were obtained and are shown in Appendix A. Notably, the absorbance peaks for the Pd^2+^ salt are absent in both spectra. This indicates the complete reduction of Pd^2+^ to Pd^0^ in the presence of NaBH_4_. The exponential decay observed in the spectra corresponds to the Mie-scattering for spherical nanoparticles. Thermogravimetric analysis (TGA) was used to determine the ratio between the organic ligand and the metal nanoparticle surface. As depicted in Appendix A, the nanoparticle consists of approximately 60% palladium and 29% ligand in addition to 11% solvent (calibrated TGA results: 68% palladium and 32% ligand). In comparison, C6-PdNPs (carboxylate) have been shown by TGA analysis (Appendix A) to consist of approximately 75% Pd and 25% ligand [40].

TEM images were taken for the newly synthesized C5-PdNPs as shown in Figure 1. Results indicate C5-PdNPs have an average core size of 3.20 ± 0.80 nm. Comparatively, the size of these nanoparticles is slightly larger than that of the C6-PdNPs, which had an average core size of 1.7 ± 0.9 nm (Appendix A) [40]. TEM images showed that both nanoparticles are spherical and well dispersed.

### 3.3. Catalytic Reactions of Allyl Benzene

The catalytic activity of two distinct nanoparticles with different ligand functionalities was tested for the reaction of allyl benzene. The reactions were performed with a 10 mol% (mole of Pd/mole over mole of allyl benzene × 100) PdNP catalyst dissolved in H_2_O at room temperature in the presence of 1 atm H_2_ (g). The ^1^H NMR spectra showed the chemical shifts for both aldehyde and allyl alcohol products indicating the oxidation of the terminal alkene group (see Appendix A, Appendix A). Table 1 shows the catalysis results obtained from the reactions of allyl benzene based on the GC-MS spectra of the resulting products.

After the reaction of allyl benzene with C6-PdNP, the analyses of reaction products using GC/MS at 6 and 24 h indicated that the hydrogenation or isomerization of allyl benzene was not the major reaction route for this reaction. Instead, the biphasic catalytic reaction under atmospheric H_2_ gas at room temperature resulted in the formation of 3-phenylpropanal (29% after 6 h and 66% after 24 h) as a major product with 3-phenyl-2-propen-1-ol (5% after 6 h and 6% after 24 h) as one of the minor products. The unusual oxidation of allyl benzene under these conditions confirmed the addition of H_2_O during the reaction. The absence of products such as 1-phenyl-2-propanone and 1-phenyl-2-propanol suggested that the addition reaction is regioselective promoting the anti-Markovnikov addition of H_2_O to the C=C bond of allyl benzene. The presence of 1-phenylprop-1-ene and propyl benzene indicated the minor occurrence of hydrogenation and/or isomerization of allyl benzene. In addition, other oxidation products including cinnamaldehyde, cinnamic acid, and benzaldehyde were observed for the 24 h reaction. The combined yields for oxidation–hydration products and isomerization–hydrogenation products were 89% and 4%, respectively.

When the concentration of H_2_ gas was diluted by using the 50:50 mixture of H_2_ and N_2_ gases, the selectivity for 3-phenylpropanal decreased significantly to 39%. The goal for this reaction condition was to increase the aldehyde selectivity by limiting the concentration of hydrogen, thus leading to a decrease in the formation of hydrogenation products. However, the reaction with diluted H_2_ gas increased the yields for overoxidation products (41%), cinnamaldehyde, cinnamic acid, and benzaldehyde, in addition to that for isomerization product, 1-phenylprop-1-ene. We also attempted to further investigate the effects of hydrogens gas by performing our reactions under aerobic conditions, which resulted in an overall impeded reaction.

A possible route for the formation of cinnamaldehyde is via the oxidation of cinnamyl alcohol (route A) as depicted in Figure 4. The generation of cinnamaldehyde is thought to be facilitated by the anionic functionality of carboxylate ligands that are present in C6-PdNPs, which may assist in the oxidation to the carbonyl analog via E2-like elimination of H. Additionally, the formation of cinnamic acid (route B) takes place by the addition of H_2_O to the carbonyl group followed by the elimination of 2 hydrogens. Small amounts of benzaldehyde would form by the hydrolysis of cinnamaldehyde in a retro-aldol-type reaction [43].

The addition of DMF as a co-solvent was detrimental to the selectivity of C6-PdNP for oxidation products, 3-phenylpropanal (12%), 3-phenyl-2-propen-1-ol (16%), and over-oxidation products (18%). Instead, the reaction in H_2_O-DMF increased the formation of isomerization-hydrogenation products, 1-phenylprop-1-ene (34%) and propyl benzene (4%).

For the inhibition of overoxidation reactions and to increase the selectivity toward 3-phenylpropanal, we investigated the catalytic activity of C5-PdNP with cationic functionality that is less susceptible to the elimination of a-hydrogen from cinnamyl alcohol. The results showed that C5-PdNP was also appreciably selective for the anti-Markovnikov Tsuji–Wacker oxidation of allyl benzene achieving an average aldehyde yield of 83% without the formation of 3-phenyl-2-propen-1-ol. Cinnamaldehyde and cinnamic acid, which are other major overoxidation products, were also absent. The isomerization and hydrogenation products, 1-phenylprop-1-ene (7%) and propyl benzene (5%), respectively, are identified as the main side products of this reaction.

As an effort to minimize the degree of hydrogenation for the C5-PdNP, the catalytic reaction was again attempted in a 50:50 mixture of H_2_/N_2_ gas. As expected, this condition led to a slight decrease in the formation of propyl benzene (2%). However, a slight decrease in the amount of 3-phenylpropanal (71%) was observed due to the formation of 1-phenylpropan-1-one (4%) and the increase in benzaldehyde (7%). Overall oxidation yields in a 50:50 mixture of H_2_/N_2_ gas, however, were nearly unchanged (85% in H_2_ gas vs. 83% in H_2_/N_2_ gas). The formation of a minute amount of 1-phenylpropan-1-one, which was absent entirely for the catalysis with C6-PdNP, hinted at the occurrence of a Pd-allyl intermediate and the nucleophilic attack by H_2_O at the benzylic site of allyl benzene (Markovnikov addition). The addition of DMF was again damaging to the selectivity of C5-PdNP for 3-phenylpropanal (14%) formation. The yields for isomerization–hydrogenation products, 1-phenylprop-1-ene (26%) and propyl benzene (5%), were significantly increased for the reaction in H_2_O-DMF.

Overall, C5-PdNP has proven to be quite effective for the anti-Markovnikov oxidation of allyl benzene producing 3-phenylpropanal in 83% yield in water at room temperature. C5-PdNP has also resisted the formation of additional oxidation products such as cinnamyl alcohol, cinnamaldehyde, and cinnamic acid observed for the C6-PdNP. It is recognized that oxidation of allyl benzene in the presence of H_2_ (g) is quite an unusual case since the Tsuji–Wacker oxidation is normally performed under aerobic conditions to afford methyl ketones.

The substrate scope of two nanoparticle catalysts, C5- and C6-PdNPs, was further examined using styrene and 4-phenylbut-1-ene. By using these substrates, we sought to maintain the interaction between the aromatic groups and the metal catalyst by using different aromatic olefinic compounds [44,45]. To our surprise, however, little to no oxidation was observed using either set of PdNPs under the same reaction condition as depicted below (Figure 5).

Instead of oxidation, the isomerization/hydrogenation (85%:14%) products were prevalent for 4-phenylbut-1-ene whereas only a minute amount of hydrogenation was observed for styrene. Although these preliminary studies narrowed the substrate scope for this reaction, some useful insight was obtained with regard to the importance of allylic hydrogens, which are speculated to play an important role in the oxidation-hydration processes. As we may note, styrene does not have any allylic hydrogens present and are also not capable of undergoing isomerization. On the other hand, both allyl benzene and 4-phenylbut-1-ene have allylic hydrogens and may undergo isomerization in the presence of H_2_ (g). However, this does not explain why no oxidation was observed for 4-phenylbut-1-ene since it meets the above criteria. A plausible explanation could be that the allylic hydrogens relative to the alkene are too far away from the benzene ring to experience additional benzylic effects and/or form a stable arene/Pd transition state. Altogether, these results point toward the potential C-H activation of allyl benzene, which may be dependent on the concentration of hydrogen.

Based on these studies, the proposed mechanism begins with the coordination of the double bond to form an η^2^ Pd-alkene intermediate (Figure 6) [46]. The anti-Markovnikov addition of H_2_O about the less hindered carbon atom can occur via one of two intermediates, branched Pd-alkyl intermediate or Pd-allyl intermediate [47,48,49]. Under our conditions, we would normally expect to form a branched Pd-alkyl intermediate due to the excess of H_2_ (g) [16]. However, this intermediate is geared toward forming the hydrogenation and isomerization products after the addition of hydrogen about either carbon center. Assuming the simultaneous addition of H_2_O and the formation of a branched Pd-alkyl intermediate, the role of hydrogen gas would become more obscure. On the contrary, if we assume that our substrate undergoes oxidative addition to our palladium center then we can reasonably account for the crucial presence of hydrogen. This alternative pathway would lead to the formation of an η^3^ Pd-allyl intermediate after the cleavage of hydrogen α to a benzene ring.

Following this step, the addition of the H_2_O to the less substituted carbon atom of the Pd-allyl intermediate would form cinnamyl alcohol. This then leads to the formation of a common branched Pd-alkyl intermediate, which can underdo β-hydride elimination at two different carbon centers. The abstraction of hydrogen α to the phenyl group would lead to the production of 3-phenyl-2-propen-1-ol (cinnamyl alcohol), which has often been observed as one of the major products for the C6-PdNPs. The removal of hydrogen α to the hydroxyl group results in an enol intermediate that can favorably tautomerize to form 3-phenylpropanal, which is a significant or major product for both PdNPs. Interestingly, the isomerization of cinnamyl alcohol to 3-phenylpropanal could be enhanced if the concentration of hydrogen is increased. Thus, the Pd-allyl pathway may account for the necessary presence of hydrogen gas to achieve significant aldehyde yields. Whatever the nucleophilic mode of addition may be, it is recognized that a common branched Pd-alkyl intermediate must be generated to produce 3-phenylpropanal and 3-phenyl-2-propen-1-ol.

## 4. Conclusions

Synthesis of water-soluble Pd nanoparticles was successfully achieved using the modified thiosulfate protocol. These PdNPs demonstrated excellent regioselectivity for the anti-Markovnikov Tsuji–Wacker oxidation of allyl benzene, via the addition of water about the less hindered carbon atom, under mild conditions using only H_2_ (g) as a secondary reagent. No other reagents such as base, salts, and organic cosolvent were added. C5-PdNPs demonstrated higher aldehyde selectivity than the C6-PdNPs (~83% vs. ~66%, respectively). C5-PdNPs were slightly more prone to hydrogenation/isomerization reactions but suppressed additional oxidation reactions. This was contrary to the results for the C6-PdNPs, which produced a significant amount of additional oxidation products. Mechanistically, the nucleophilic addition of H_2_O took place at the Pd-allyl intermediate with the necessary presence of benzylic hydrogens. The addition of water to the less substituted carbon is a technologically important process because this reaction can be used to form primary alcohols and related derivatives. Since the regioselective addition of H_2_O by water-soluble PdNPs is operated under mild conditions without any additional reagent, this discovery might turn out to be useful for the practical applications of colloidal nanoparticle catalysts with further optimization expanding the scope of substrates.

## Data Availability

Not applicable.

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
