# Peer review of "Water-Soluble Pd Nanoparticles for the Anti-Markovnikov Oxidation of Allyl Benzene in Water"

_nanomaterials, 2023, doi:10.3390/nano13020348_

Round 1

Reviewer 1 Report

In this paper, the authors reported the synthesis of two water-soluble palladium nanoparticles capped with 5-(trimethylammonio)pentanethiolate (C5-PdNP) and 6-(carboxylate)hexanethiolate (C6-PdNP) for oxidation catalyst of allyl benzene to form 3-phenylpropanal.

The characterization of C5-PdNP and C6-PdNP was performed suitably by various measurements such as 1H NMR, UV-Vis spectra, TGA, and TEM.

The catalytic assays of C5-PdNP and C6-PdNP were performed by analyzing the isolated products via 1H NMR and GC/MS.

They reported that these PdNPs especially C5-PdNP showed excellent regioselectivity for the anti-Markovnikov Tsuji-Wacker oxidation of allyl benzene.

I think that the results reported in this paper may be useful to develop of the field of colloidal nanoparticle catalyst.

Minor comments

Line 243

   9 % → 6 %       ?

Author Response

Summary: In this paper, the authors reported the synthesis of two water-soluble palladium nanoparticles capped with 5- (trimethylammonio)pentanethiolate (C5-PdNP) and 6-(carboxylate)hexanethiolate (C6-PdNP) for oxidation catalyst of allyl benzene to form 3-phenylpropanal. The characterization of C5-PdNP and C6-PdNP was performed suitably by various measurements such as H NMR, UV-Vis spectra, TGA, and TEM. The catalytic assays of C5-PdNP and C6-PdNP were performed by analyzing the isolated products via H NMR and GC/MS. They reported that these PdNPs especially C5-PdNP showed excellent regioselectivity for the anti-Markovnikov Tsuji-Wacker oxidation of allyl benzene. I think that the results reported in this paper may be useful to develop of the field of colloidal nanoparticle catalyst.

Response: We thank the Reviewer for his/her time and encouraging words.

Remark #1. (Line 243) 9 % → 6 % ?

Response: We thank the reviewer for noting our mistake. The corresponding change was made in the revised manuscript.

Reviewer 2 Report

The manuscript describes the catalytic activity and selectivity of two water-soluble palladium nanoparticles coated with ligands which are studied employing the catalytic reaction of allyl-benzene. The results that the authors report are significant and demonstrate that the regioselective oxidation of allyl-benzene occurs at room temperature at atmosferic pressure and pure water via Tsuji-Wacker-type oxidation (conventionally Markovnikov oxidation of terminal alkenes to their respective ketones). Water-soluble Pd nanoparticles catalyze the anti-Markovnikov oxidation with a yield of about 83%. The catalytic results of other aromatic alkenes suggest that the presence of benzyl hydrogen is a key to the formation of a Pd-p-allyl intermediate and anti-Markovnikov addition of H20. The authors describe and highlight the characteristics of the particles mechanisms also with the use of sophisticated analytical instruments. Certainly the results obtained are significant and flattering for the development of science even if they are not  such as to procure a Nobel prize. In my opinion due to its very significant content the manuscript can be taken into consideration for publication in Nanomaterials.

Author Response

Summary: The manuscript describes the catalytic activity and selectivity of two water-soluble palladium nanoparticles coated with ligands which are studied employing the catalytic reaction of allyl-benzene. The results that the authors report are significant and demonstrate that the regioselective oxidation of allyl-benzene occurs at room temperature at atmosferic pressure and pure water via Tsuji-Wacker-type oxidation (conventionally Markovnikov oxidation of terminal alkenes to their respective ketones). Water-soluble Pd nanoparticles catalyze the anti- Markovnikov oxidation with a yield of about 83%. The catalytic results of other aromatic alkenes suggest that the presence of benzyl hydrogen is a key to the formation of a Pd-p-allyl intermediate and anti Markovnikov addition of H20. The authors describe and highlight the characteristics of the particles mechanisms also with the use of sophisticated analytical instruments. Certainly the results obtained are significant and flattering for the development of science even if they are not such as to procure a Nobel prize. In my opinion due to its very significant content the manuscript can be taken into consideration for publication in Nanomaterials.

Response: We thank the Reviewer for his/her time and highly supporting words.

Reviewer 3 Report

This paper describes the use of Pd nanoparticules for the anti-markovnikov oxydation of allylic/aromatic derivatives.

The work is well described and conducted, and the results are convincing.

My main concern relates with the synthetic usefulness of this work. Indeed, it seems that altough the oxydation process does not requires any co-oxidant (such as AgNO3 or oxygen), the couterpart is that its scope appears (yet) as very limited. Attempts to oxydize different substrates where the aromatic/alkene moieties are distant proved to be unsuccessful.

In order for this paper to be accepted for publication, it seems that the authors should evaluate the efficiency of their catalyst on a larger panel of  substrates, bearing diverse functional groups.This will: 1) reinforce this paper, and 2) some functional groups may exert an orientation / activation effect that could relax the constraint of having aromatics and alkenes closely associated.

Author Response

Summary: This paper describes the use of Pd nanoparticules for the anti-markovnikov oxydation of allylic/aromatic derivatives. The work is well described and conducted, and the results are convincing. My main concern relates with the synthetic usefulness of this work. Indeed, it seems that although the oxydation process does not requires any co-oxidant (such as AgNO3 or oxygen), the couterpart is that its scope appears (yet) as very limited. Attempts to oxydize different substrates where the aromatic/alkene moieties are distant proved to be unsuccessful. In order for this paper to be accepted for publication, it seems that the authors should evaluate the efficiency of their catalyst on a larger panel of substrates, bearing diverse functional groups. This will: 1)reinforce this paper, and 2) some functional groups may exert an orientation / activation effect that could relax the constraint of having aromatics and alkenes closely associated.

Response: We thank the Reviewer for his/her time and supporting words. We agree with the reviewer that the inclusion of additional catalysis results for other substituted aromatic compounds with different functional groups and/or aliphatic alkenes would improve the overall quality of the manuscript and reinforce the new findings for the broader audience of scientific communities. However, the current manuscript is already 13 pages long and discusses the influence of other structural effects of aromatic substrates (styrene vs. allyl benzene vs. 4-phenylbut-1-ene (homoallylic benzene)), we would like to report additional catalysis studies soon as a complementary investigation after reaching reasonable conclusions.